# Molecular Identification of Asian Hornet *Vespa velutina nigrithorax* Prey from Larval Gut Contents: A Promising Method to Study the Diet of an Invasive Pest

**DOI:** 10.3390/ani13030511

**Published:** 2023-02-01

**Authors:** Kirsty Stainton, Sam McGreig, Chris Conyers, Sally Ponting, Lee Butler, Paul Brown, Eleanor P. Jones

**Affiliations:** 1Fera Science, The National Agri-Food Innovation Campus, Sand Hutton, York YO41 1LZ, UK; 2Oxford Nanopore Technologies, Gosling Building, Edmund Halley Road, Oxford Science Park, Oxford OX4 4DQ, UK; 3School of Natural and Environmental Sciences, Agriculture Building, King’s Road, Newcastle upon Tyne NE1 4LB, UK

**Keywords:** *Apis mellifera*, apiculture, *Vespa velutina*, invasive species, diet metabarcoding

## Abstract

**Simple Summary:**

The potential impact of the invasive Asian hornet on honey bee and insect biodiversity is not well understood. This study tests a method to identify what species the Asian hornet preys upon. We took the stomach contents of Asian hornet larvae and identified the prey items within those using DNA methods. The method was successful and many prey items were identified. Although the number of nests and larvae tested were small (and the results therefore had to be treated with caution), they gave an interesting snapshot of the prey items captured by *V. v. nigrithorax* in the UK. The method has great potential to gather further information on the diet of the Asian hornet as it expands into new areas.

**Abstract:**

The Asian hornet, *Vespa velutina nigrithorax* (Hymenoptera: Vespidae), is an invasive hornet that was accidentally introduced into Europe in 2004. It mainly preys on other invertebrates and arthropod species, and often targets honey bee (*Apis mellifera*) colonies. The introduction of these hornets may damage indigenous fauna and apiculture. Knowledge of *V. velutina* prey preference and the species composition of their diet is relatively limited. In this study, we assessed methodologies for the molecular identification of prey using dissected larvae from destroyed nests. Ten larval samples were taken from five nests in areas where the hornets had not yet established: two from the Channel Islands and three in the mainland UK. DNA was extracted from the gut contents and sequenced and analysed by metabarcoding with Oxford Nanopore Technologies’ Flongle and MinION devices. Numerous taxa were detected in each larval sample with the species composition varying by individual and by nest. Between 15 and 26 species were found per nest, with wasps (*Vespula* spp.), spiders, honey bees and blow flies being the most abundant taxa. These results demonstrate that metabarcoding larval gut contents can be used to study the Asian hornet diet and give a first snapshot of the prey items captured by *V. v. nigrithorax* in the UK. This method could be used for future large-scale testing of the gut contents of hornet nests, in order to provide a greater insight into the foraging behaviour of this predator across Europe and elsewhere.

## 1. Introduction

The yellow-legged Asian hornet, *Vespa velutina nigrithorax*, was accidentally introduced into the south-west of France in 2004 [1] and has now spread across three quarters of France [2]. *V. v. nigrithorax* has since colonised Spain, Portugal, Belgium, the Netherlands, Switzerland, Luxembourg, Italy [3,4,5] and Germany [6]. In the United Kingdom, it has been found sporadically since 2016 [7], but active management is currently being undertaken to prevent it from establishing in the UK, which appears to be successful [8]. This invasive hornet is a cause for concern due to its insectivorous nature and the potential impact it has on native arthropod species and managed honey bee (*Apis mellifera*) colonies [9]. While the adult hornets predominately sustain themselves on carbohydrate sources from nectar, larval secretions or ripe fruits [10], they must continuously forage protein sources for the developing brood. This protein comes in the form of a ‘flesh pellet’: a small piece of proteinaceous material that the adult carves out from the protein rich thorax of a captured arthropod, or from carrion [11]. Anecdotal information suggests that hornets are opportunistic predators that will exploit any protein source they encounter, including fish or meat markets. The ecological impact of *V. v. nigrithorax* in Europe is not yet fully understood, as there is insufficient data on native arthropod populations before and after colonisation to determine its effects, but it is known to negatively affect honey bee survival, at least in some locations [9]. Carisio et al. [12] assessed the ecological impact of *V. v. nigrithorax* on a native hornet (*Vespa crabro*) and wasps (*Vespula* spp.) as it colonised NW Italy, but there have been no systematic studies of the many species potentially impacted across a wide area.

By studying the diet of *V. v. nigrithorax*, a better understanding of the potential impact it may have on the local fauna and on beekeeping can be ascertained, as well as a better understanding of its natural history to help guide management by: (a) determining likely locations to detect foraging hornets, and (b) to create management plans to better protect biodiversity and apiculture as necessary. To date, predation studies of the Asian hornet in Europe have primarily been from capturing foraging hornets returning to the nest with flesh pellets and identifying the pellet, but there is a study that used DNA metabarcoding to identify prey from samples of adult jaws and guts, in addition to larval faecal pellets [13]. Studies by Perrard et al. [14], Villemant et al. [15], and Rome et al. [3] collected flesh pellets from captured hornets returning to their nests and identified these pellets morphologically. These studies found the honey bee to be a major dietary component (85% and 37% of pellets, respectively), along with Diptera, carrion and other arthropods. Although these studies have provided a valuable insight into hornet foraging behaviour, morphological identification from the butchered remains of an arthropod thorax is not always possible and can be difficult to resolve to the species or genus level. An alternative approach to identification is to use methods that rely on DNA barcoding [16]. This approach was used in Rome et al. [3], who identified 2151 individual prey pellets using both morphology and DNA barcoding.

However, capturing individual hornets to confiscate prey pellets is a time consuming and potentially hazardous activity; Perrard et al. [14] took 6 days to collect 235 pellets. Alternative samples to identify prey items can be used, as demonstrated by Verdasca et al. [13], who used DNA metabarcoding to identify prey from samples collected from adult jaws and stomachs and larval faecal pellets in Portugal. A similar approach has been taken by LeFort et al. and Schmack et al. [17,18], who used metabarcoding to identify the diet from faecal samples taken from within the nests of invasive *Vespula* species in New Zealand. The advantage of faecal samples is that they can be taken from the nest; however, the DNA from the prey species will be heavily degraded. In the current study, we use larval gut content samples as a proof of concept to see if they can be used to reconstruct the Asian hornet diet.

Metabarcoding mixed samples has been extensively used for dietary studies, including on insects (e.g., [17,18]) and other arthropods [19]. Although a powerful method, the interpretation of the data can be ambiguous, particularly over the interpretation of sequence read counts and whether they can be used to quantify the biomass of a species/ taxon in the diet (e.g., [20,21]). Within the present study, we take the presence of a species/taxon above a given threshold of read counts in the data as being a true positive (i.e., it is actually present in the sample). A positive species presence was determined by identifying the taxon with the most reads assigned to it in a sample, and subsequently setting a threshold based on 1% of this value—an approach which may scale better with diverse samples. Despite the ambiguity of interpretation, we also include the sequence read counts, as we believe they are likely to contain some biologically meaningful information, e.g., a species that comprises 0.1% of the reads is generally likely to have comprised a smaller component of the larval stomach contents than a species that comprises 60% of the reads.

To improve our understanding of the feeding habits of *V. v. nigrithorax*, we developed a molecular method for the identification of hornet prey through dissecting the gut contents from developing larvae from five nests and metabarcoding them. Larval gut content samples have the advantage of being simple to recover when nests have been collected, and they represent a less degraded sample type than faecal pellets. Initial testing was performed on a single nest from Jersey sequenced on a Flongle Flow Cell from Oxford Nanopore Technologies (ONT) as a proof of concept. As initial testing provided good depth of coverage, follow-up testing on four further nests was performed using a MinION flow cell.

We discuss the success of these methods and their potential future application across a variety of landscapes and timeframes to gain a better insight into the impact of this invasive predator. The results from the study are small scale (a relatively small number of samples, from five nests) and are unlikely to be fully representative of the hornet’s diet should it establish in the UK. With those caveats in mind, we explore the implications of the findings for UK apiculture and biodiversity.

## 2. Materials and Methods

### 2.1. Collection of Hornet Samples

Five nests from the British Isles were analysed. These were nests from Alderney, Channel Islands (found and destroyed in October 2016), from Tetbury, Gloucestershire (September 2016), Woolacombe, Dorset (September 2017), Jersey, Channel Islands (August 2019) and Gosport, Hampshire (September 2020) (Figure 1). The Alderney and Jersey nests were stored at −15 °C and shipped rapidly to Fera Science in a cool box, while the UK mainland nests were destroyed with an insecticide and shipped rapidly to Fera Science in a cool box and stored at −20 °C immediately upon arrival. Figure 1 shows a location map for all of the nests.

Individual larvae were removed from the nests at Fera Science soon after recovery and placed into individual containers (either 15 mL Falcon tubes or 1.5 mL Eppendorfs) and frozen at −15 °C (short term) or −50 °C (long term). Ten larval samples were used from each nest. However, prey DNA was not successfully amplified for all samples, with the Alderney nest failing in four out of ten samples and the Gosport nest failing in seven out of ten samples (Table 1). It is likely that these failures will be due to inadequate storage of the samples (samples being left too long before being frozen, or repeated freeze/thaw cycles).

### 2.2. Sampling of Gut Contents, DNA Extraction and PCR

Gut contents were carefully dissected from the largest larval stages into Eppendorf tubes for each individual, taking care to avoid any larval gut tissue. A sub-sample from the total gut contents for each larva was DNA extracted using a Qiagen Blood and Tissue Kit using the manufacturer’s protocol. For the Jersey samples, PCR was performed using the general invertebrate primers LCO22me (5′-GGTCAACAAATCATAAAGATATTGG) & HCO700dy (5′-TCAGGGTGACCAAAAAATCA) from [22] and CI-J-2183 (5′-CAACATTTATTTTGATTTTTTGG) & L2-N-3014 (5′-TCCAATGCACTAATCTGCCATATTA) from [23] in 20 µL reactions with 1–10 µL DNA, 200 µM each primer and 10 µL BioXACT short DNA polymerase (Bioline). The volume of DNA added was determined by whether the PCR amplification was successful; for some samples, increased DNA led to greater PCR success. BSA (bovine serum albumin) (0.5 µL of 20 mg/mL) was added to each reaction. For the Tetbury, Woolacombe, Alderney and Gosport samples, a ~650 nucleotide fragment of the mitochondrial Cytochrome Oxidase I (COI) gene was amplified using the LC01490 (5′ GGTCAACAAATCATAAAGATATTGG) & HC02198 (5′-TAAACTTCAGGGTGACCAAAAAATCA) primers [24]. All PCRs were run using PCR MasterMix MyFi^®^ Taq 2X Master Mix (BioLine, Meridian Biosciences, Cincinnati, OH, USA), a high-fidelity proof-reading enzyme. PCR amplicons were run on a 1% agarose gel stained with ethidium bromide to confirm correct amplification. For the Gosport samples, only 3/10 samples successfully amplified, and although they were subsequently sequenced and discussed, it should be noted that this is an insufficient sample size from which to draw any broad conclusions about the nest.

### 2.3. Metabarcoding

For the Jersey samples, the purified PCR products underwent library preparation and were run on the Oxford Nanopore Technologies Flongle flow cell (R9.4.1). Library preparation was carried out according to the Oxford Nanopore Technologies Native barcoding amplicons protocol using a Ligation Sequencing Kit SQK-LSK109 and 20 individual barcodes from the expansion kits EXP-NBD104 and EXP-NBD114 (12 unique barcodes supplied in each kit). Briefly, the ends of the DNA amplicons were prepared by end-repairing enzymatically for adapter attachment. Native barcodes were attached to the DNA ends by blunt end ligation. The individual samples were normalised to equal quantities of DNA, as quantified on a Qubit 4 Fluorometer (ThermoFisher) and pooled. Following the pooling of the samples, sequencing adapters were ligated to the barcodes and the subsequent pool was prepared for loading by the addition of a sequencing buffer and loading beads (Bead ratio 0.367, 1 µL beads: 13.5 µL sequencing buffer: 5.5 µL library). The prepared pool was loaded onto a primed Flongle flow cell (R9.4.1) and sequenced.

For the Tetbury, Woolacombe, Gosport and Alderney samples, the amplicons generated in the first round of PCR were cleaned using AmpureXP beads (Beckman Coulter, High Wycombe, UK) to remove excess primers and PCR reaction constituents that could interfere with the next round of barcoding PCR. The cleaned amplicons then underwent a second round of PCR to add an individual barcode ‘label’ which allows the multiple samples to be mixed in one pool and sequenced in one run on the MinION. Equal amounts of each barcoded sample were added to make the final pool, which was then sequenced on a MinION SpotON flow cell (R10.3) following the PCR SQK-LSK109 protocol to end repair and dA tail the fragmented DNA, ligate adaptors, and load onto the flow cell (Oxford Nanopore Technologies, Oxford, UK).

### 2.4. Bioinformatics Analysis

For the MinION dataset (Tetbury, Woolacombe, Gosport and Alderney samples), 29 samples were sequenced on an Oxford Nanopore MinION sequencer, using an R10.3 flow cell, over 48 h. Reads were basecalled with Guppy (version 5.0.11) in high accuracy mode. In total, 885,593 reads were generated, equating to roughly 631 Mb, with an average quality score of 11.6 (~6.9% error rate) and a median read length of 705. A PCR and barcode negative were included, generating 128 and 61 reads, respectively.

For the Flongle dataset (Jersey samples), nine samples were sequenced on an Oxford Nanopore Flongle adapter on a MinION device, using an R9.4.1 flow cell, over 48 h. Reads were basecalled with Guppy (version 5.0.11) in high accuracy mode. In total, 70,375 reads were generated, equating to roughly 56 Mb, with an average quality score of 10.2 (~9.5% error rate) and a median read length of 758.

Reads were trimmed with Cutadapt [25], and then any reads which were too long (greater than 750 bases) or short (fewer than 550 bases) for the expected amplicon size were removed. Subsequent filtering was applied to remove any reads which had a Phred quality score of less than 10 for MinION sequences and 7 for Flongle sequences. A custom COI database was built from sequences obtained from Genbank, and individual reads were subject to a BLASTn (MegaBlast) [26] search against this database. The resulting reads were filtered so that only matches to sequences in the database that had a percentage identity of at least 85% and an alignment length of at least 80% were included. Of these hits, only the hits with a bitscore within 3% of the highest scoring hit were included in the final set of filtered reads. Finally, a lowest common ancestor approach was applied to the dataset, where if at least 75% of the assignments agreed, that taxonomic label was applied to the read. Otherwise, the next highest rank was considered, and the process was repeated until a label was assigned to the read. Where a lowest common ancestor could not be agreed upon, reads were assigned as ‘Unresolved’. For each sample, the taxa with the highest number of reads was selected, and then any taxa with a number of reads greater than or equal to 1% of this number was also selected. The reads associated with these taxa were extracted and used to build a higher accuracy consensus sequence with Oxford Nanopore’s Medaka tool [27] (Oxford Nanopore Technologies Ltd., Oxford, UK, 2018).

The output of the analysis was a matrix, detailing the number of reads assigned to each taxon per sample. However, there will be a degree of ‘noise’ in the results due to barcode misassignment, MinION error rates (5–15% per read is typical for Nanopore sequencing platforms using older kits and chemistries) and low-level background contamination. We therefore applied a threshold of 1% of the total number of reads assigned to the taxa with the most reads for that sample; all taxa above this threshold were accepted as ‘true’ results, while those below this threshold were excluded from further analysis. This approach allows for the inclusion of smaller but biologically relevant clusters of reads in the analysis. A further issue with the output was that single species were occasionally split across multiple taxonomic ranks in the resulting matrix (i.e., appearing at the species level, at the genus level, and at the family level). To rectify this issue, once this threshold was applied, consensus sequences for the remaining taxa were reviewed manually, and any taxa that were assigned to a higher-level rank (e.g., family level) were assessed and assigned to the species or genus level where possible. Where a group of sequences could not be resolved to a particular genus or species, but there is confidence that this represented a single species, the taxa was kept at a single matrix entry at the higher level (e.g., for the family Acrididae). Errors in the DNA databases led to some obvious misassignments, such as several assignments that identified as an insect when they were of bacterial origin. These became apparent after the manual examination of the sequences, and they were assigned to ‘uncultured bacterium’. The filtered matrix was used to generate subsequent plots.

## 3. Results

### 3.1. Data Quality

Ten samples were taken from each of the nests (nine samples from Jersey), but prey DNA was only successfully amplified from some samples (Table 1). There was 100% successful amplification from the Jersey, Tetbury and Woolacombe samples, but only 60% and 30% success from Alderney and Gosport, respectively (Table 1). This low success rate may be due to poor initial storage of the materials or to repeated freeze-thaw cycles, which partially degraded the DNA templates. The average number of DNA sequence reads per larva was consistent with what was expected from the different platforms; the Jersey nest has fewer reads per larva (an average of 5827 reads per larva), as a Flongle flow cell has a lower expected yield than a MinION flow cell, where the number of reads obtained was 3- to 4-fold higher (Table 1). Despite using a different platform, the number of species per nest was consistent between nests, and there were between 15 and 26 prey species recorded across the five nests (Table 1). Although one gut from the Tetbury nest provided reads for only a single species (*Dryomyza anilis*), the average number of different species detected per gut across all nests was 6.7, with the highest number of species detected in a single gut being 13 from a sample from Gosport. There was some contamination of the samples with reads from the host itself; one from Tetbury yielded reads exclusively from *V. velutina*, while three others from Tetbury, two from Alderney and one from Woolacombe had a small number of host reads but also contained reads from other species.

Metabarcoding data can be presented in terms of incidence, i.e., how many gut samples test positive for a given species, or in terms of abundance, i.e., the amount of reads per species for each individual or nest. However, read data is not quantitative, and two different Oxford Nanopore Technologies devices were used, so no quantitative statistical analysis has been performed, rather the data provide a qualitative observation of prey consumed. The error rates reported in this study reflect older chemistries from Oxford Nanopore Technologies; current raw read accuracy is now higher and exceeds Q20. In addition, the data presented were not basecalled in super accuracy ‘SUP’ mode, which would have improved the accuracy further.

### 3.2. Prey Species Found

In total, thirty-eight larval gut samples were analysed from five nests (summarised in Table 1).

*Vespula* spp. (European and German wasps) was consistently the most abundant prey item in all nests, as measured by total reads across all samples (Table 2 and Appendix A, Figure 2), as the species found most often in larvae (26 out of 38 samples) and as the only species recorded in all nests (Table 2 and Appendix A, Figure 2). They also accounted for the highest proportion of read counts in the Tetbury, Alderney and Gosport nests (Table 2 and Figure 2).

The honey bee *Apis mellifera* was also found frequently, being present in 25 of the 38 larva sampled and found in four of the five nests (Table 1 and Appendix A). The absolute abundance of honey bee reads across all samples was quite low (Appendix A).

The blow flies (Family: Calliphoridae) were a common prey item across all nests, and were the group with the highest read count in the Jersey nest (Figure 2). These were from the genera *Calliphora* spp., *Lucilia* spp. and *Pollenia* spp., with *Calliphora* the most abundant. Common species were *C. vicinia* and *C. vomitoria*, although they were not found in all nests (Appendix A). *Lucilia* spp. and *Pollenia* spp. DNA were present at a low read number in most of the nests. Hoverflies (family: Syrphidae) were present in three of the nests, represented by a range of species: *Eristalis* spp., *Scaeva pyrastri, Sericomyia silentis, Eumerus strigatus* and *Syrphus vitripennis.*

Spiders were also commonly found, being present in four out of five nests and contributing to 10% or more of the read counts in two of the five nests (Table 2 and Appendix A). These were from three species: *Araneus diadematus, Metellina segmentate, Zygiella* spp. (likely *Z. x-notata*; Table 2).

Some groups or species were only present in one or two nests, which may represent a local source of these species but could also be an artefact of the low numbers of samples in the study. Examples of this are *Dryomyza anilis* at the Tetbury nest, found in six out of ten larvae, peaking at 21,574 reads in one gut (Appendix A). *Dryomyza* spp. larvae are associated with rotting carrion and faeces [28], and the Tetbury nest may have been close to a population of these flies. Perhaps co-incidentally, *Erinaceus europaeus*, the European hedgehog, was also found in a single larva from the Tetbury nest, likely scavenged from carrion.

A number of genera of hover flies were detected. These were not resolved to species level, however at least some of these genera contain both species that are common and abundant and species that are in decline or are rare in the UK (e.g., *Eristalis abusivus, Eristalis horticola, Eristalis cryptarum, Eumerus funeralis Eumerus strigatus, Eumerus sabulonum* [29]). Another exclusive group is the Tachinid flies (a family of parasitoid flies), three species of which were found across five Jersey samples and one Alderney sample. No reads were found from bumblebee or solitary bee species.

The filtered read count data for all taxa per sample can be found in Appendix A.

## 4. Discussion

### 4.1. Method Performance

Within this study, DNA sequencing tools were used to identify the prey remains within the stomach contents of Asian hornet, *V. v. nigrithorax*, larvae. The method gave a high yield of arthropod DNA reads, and it was biologically likely that these came from prey remains. Similar approaches have been used on faecal samples, e.g., a metabarcoding dietary analysis done on invasive *Vespula* species [17,18]. It is probable that the DNA in faecal samples is more degraded than that in gut contents, as it has been further processed by the insect digestive system, and gut contents may be more representative of the prey taken. However, frass samples may be easier to preserve and transport to the laboratory than larval samples.

Compared to other studies of the Asian hornet diet which captured foraging hornets [3,14], the approach described here required a relatively low amount of effort to collect samples. However, a possible advantage of collecting prey pellets, rather than metabarcoding, is that it gives an absolute count of numbers of individuals of each prey species taken during the time spent capturing hornets. Metabarcoding results report DNA sequence read counts, which are often inferred to approximate to the relative amount of biomass from each prey item in the sample. However, biases will arise from a range of sources [20], including how well DNA is extracted from each organism, how quickly different species and life stages are digested in the gut, and primer biases in which species amplify better in the PCR stage [30]. These will distort the ratios of read counts per species in the final data, and some taxa may be missed completely. A limitation of this dataset is that many prey items are not resolved to species, but rather identified to the genus or family level. The principal causes for this are likely to be the lack of reference sequences for specific species and the DNA sequence used being too short to confidently resolve to species.

This data set is confined to a small sample size and restricted in time and space. Results may vary depending on the location of the nests in the local landscape (nests close to apiaries may have different foraging habits than those close to a farm, for example) as well as across a wider geographical range (prey species will be more/less abundant in different parts of the country and in different landscape types), the time of year (which affects relative abundance of each prey species and flowering of plants which attract pollinators), and the fact that adult hornets are opportunistic and may exploit protein sources as yet unidentified. These nests are also recent arrivals in a landscape that is not competing with other *V. v. nigrithorax* nests, which may affect hornet foraging behaviour compared to an area where there is greater intraspecific competition. Discussions of prey results should be interpreted with these limitations in mind.

### 4.2. Prey Results

The results came from relatively few samples per nest (up to ten larvae, out of a typical nest population of 3500+ individuals), all collected in autumn. However, some groups were consistently present across multiple nests, and multiple larvae within the nests (e.g., wasps, honey bees, spiders, and blow flies).

Consistent among samples was the presence of *Vespula* spp. (wasps) and honey bees (discussed further in Section 4.3 below). These may be an attractive target for hornets as they come and go from static colonies in large numbers. Blow flies were commonly present in the larval diet, however the data cannot show whether these were from blow fly larvae (associated with carrion and refuse) or adults (which are attracted to flowers such as brambles and ivy, as well as fresh faeces and Stinkhorn fungus [31]). Anecdotally, Bee Health inspectors observed Asian hornets hunting over flowering ivy, which may be the source of some of the blow flies.

The relative abundance of spiders in the diet was interesting, as they were not found to be a large component of the diet in previous studies [3]. All three species (*Araneus diadematus, Metellina segmentate, Zygiella* spp. (likely *Z. x-notata*)) are orb-weavers in terms of web design, very large bodied, conspicuous in their colouring, and they frequently reside in the middle of their webs, advertising their presence. Many flying insects are attracted to the vibrant colours of flowers and spiders exploit vivid colouration for use as sensory traps [32]. These factors may make them vulnerable to predation by Asian hornets. Furthermore, all three spider species are at their largest body size in late summer to early autumn, which may contribute to their frequency in the data [33].

The dietary results are likely to reflect seasonality in prey availability. All nests were captured from August onwards (Table 1), and it is likely that other invertebrates will dominate the diet earlier in the year before wasps (and possibly spiders) become abundant. Ideally, seasonal changes in hornet diet should be assessed by nests captured earlier in the year; however, this presents logistical challenges, as nests are harder to find before hornet numbers have built up. Certainly, within the UK, all nests were detected and removed as early as possible in the year.

### 4.3. Possible Impact on Apiculture

The Asian hornet is generally considered to have a detrimental impact on honey bees [9,11]. In France, hornet nests can reach a population of up to 13,000 individuals [34] at the height of the season (although UK nests so far have produced fewer hornets [7,8]), so many hornets can intensively prey upon any honey bee apiary situated near the nest. Each honey bee colony contains tens of thousands of individual honey bee adults and larvae, and as European honey bee colonies have not co-evolved alongside *V. v. nigrithorax*, they have poorly adapted defence behaviours to protect their colonies against foraging hornets [35]. In some parts of France, predation by hornets is so intense that honey bee colonies can collapse due to a combination of a reduced population of honey bees through direct predation and the cessation of normal foraging activity; a response to high levels of hornet predation by the bees [9].

Honey bees were abundant but were not the major constituent of the larval diet in this study. This contrasts with the largest equivalent study [3] using directly sampled prey pellets identified to species (which gave an accurate count of prey items), where the single largest number of prey pellets were honey bees at 38.1% (820 out of 2151). Meanwhile, a smaller study on 10 adult jaws and guts, and 12 larval faecal pellets [13], found that 75% of reads were represented by the honey bee. In both locations, honey bees appeared to be the major diet component. It is possible that the hornets in France and Portugal preyed more readily on honey bees, as they had higher intraspecific competition with other Asian hornet nests, unlike in the UK, where Asian hornets are not established. However, we found considerable intra-nest variability in prey taken, although comparisons between areas should be made with caution considering the low numbers of nests sampled in all three studies.

### 4.4. Low Abundance and Vulnerable Species

A number of genera of hover flies were detected but could not be resolved to species. As these genera contain species that are in decline or rare in the UK, identification of species is highly important to be able to use these metabarcoding data to inform the risks to biodiversity. As discussed above, this could be resolved by better DNA reference data and longer DNA sequence reads.

The surprising absence of DNA from any bumble bees or solitary bee species in the data may be a result of the prey availability at the time of year when the nests were collected, a function of the nest locations, or an artefact of the low numbers of samples in this study, particularly if Asian hornets do not take these prey in large numbers. Small numbers of bumble bees and solitary bees were found as prey in [3]. Further studies to increase the sample size (nests and individuals) across seasons could clarify this.

Even where they do not compose a major component of the Asian hornet diet, these vulnerable species may still be impacted should the Asian hornet become established in the UK in large numbers. Opportunistic predation by Asian hornets could pose a problem, particularly in areas where honey bees or wasps (which appear to form a major component of the diet) are less abundant.

### 4.5. Applications

Where Asian hornet nests are detected and removed as part of control or monitoring strategies, the methods described here offer a relatively simple way to examine the prey taken by the nest. This method could be applied to other nest building wasp and hornet species. Further studies that expand the dataset would provide data on the comparative abundance of prey species in the diet and inform how foraging preferences may change in time and space and how these relate to local habitats and prey availability, similar to what has been done for invasive wasps in New Zealand [17,18].

Current management efforts in the UK rely on rapidly locating and removing hornet nests where the presence of hornets trigger search efforts and the flight lines of foraging hornets are used to triangulate the nest location. Linking the gut contents data with the location of the nest and the time of year across a large number of nests would allow us to gain a better understanding of hornet foraging habits and better guide efforts to find hornets in the landscape. Therefore, defining their favoured prey choice at different times of the year has a practical application for pest management. In ecologically sensitive areas, DNA barcoding would allow us to determine if hornets are negatively impacting rare species.

## 5. Conclusions

This study is a snapshot of the diet of *V. v. nigrithorax* as it enters new territory using DNA metabarcoding. Uptake of this method to identify prey in nests discovered around the UK and Europe at different times during the season could contribute to a larger understanding of this invasive predator and its impacts on native species. The method can help direct efforts to find foraging hornets in the landscape, which allows for the triangulation and destruction of nests in countries where active surveillance and control of this damaging pest is ongoing, such as the UK.

## Figures and Tables

**Figure 1 animals-13-00511-f001:**
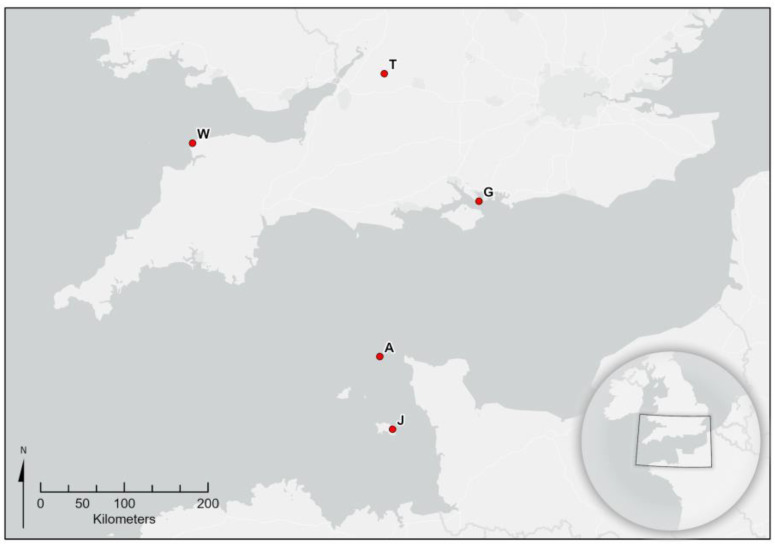
Map of Southern England and the Channel Islands showing the location of each nest used in the study. T: Tetbury, Gloucestershire, W: Woolacombe, Devon, G: Gosport, Hampshire, A: Alderney and J: Jersey.

**Figure 2 animals-13-00511-f002:**
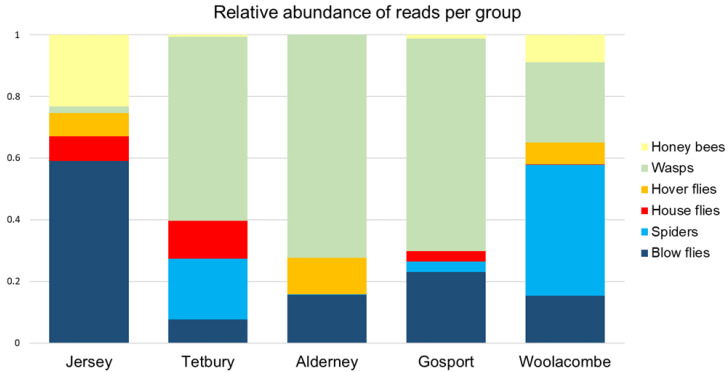
Relative abundance of the most common prey taxa presented as a proportion of all reads per nest. Taxa have been aggregated into larger groups; a more detailed taxon list can be found in Table 2.

**Table 1 animals-13-00511-t001:** List of the nests used with the approximate nest location (to preserve the anonymity of landowners), month the nest was captured and destroyed, the number of samples that were successfully sequenced (N. Seq), the average number of reads per larvae, the number of different species DNA found among the samples analysed, and the most abundant taxon found as a percentage of total reads per nest and the percentage of total reads per nest represented by *Apis mellfera* (honey bee).

Site	Grid ref. (Lat. Long.)	Month, Year Found	N. Seq.	Mean Reads per Sample	Species per Nest	Most Abundant Taxon	Mean% Honey Bee Seq
Jersey	49.2, −2.1	August, 2019	9	5827	26	Blow fly	16.5
Tetbury	51.64, −2.15	September, 2016	10	15,497	20	Wasp	0.33
Alderney	49.71, −2.20	October, 2016	6	22,194	15	Wasp	0
Gosport	50.79, −1.14	October, 2020	3	17,433	16	Wasp	0.98
Woolacombe	51.17, −4.21	September, 2017	10	18,699	15	Spider	7.3

**Table 2 animals-13-00511-t002:** List of common taxa found for each nest analysed and the relative percentage of sequencing reads assigned to each taxon for that nest.

Group	Species	Jersey	Tetbury	Alderney	Gosport	Wool.
Honey bee	*Apis mellifera*	20.5	0.3	0.0	1.2	7.3
Asian hornet	*Vespa velutina*	0.0	19.0	0.0	1.1	0.0
Wasp	*Vespula* spp.	1.9	34.3	66.6	65.9	21.6
Blow fly	*Calliphora* spp.	32.7	0.4	8.7	0.0	3.0
Blow fly	*Lucilia* spp.	15.3	1.5	5.8	21.0	0.0
Blow fly	*Pollenia* spp.	3.2	2.5	0.0	1.0	9.8
Blow fly	*Protocalliphora* spp.	0.9	0.0	0.0	0.0	0.0
Hover fly	*Eristalis* spp.	0.7	0.0	1.0	0.0	5.9
Hover fly	*Eumerus* spp.	1.1	0.0	0.0	0.0	0.0
Hover fly	*Scaeva* spp.	0.0	0.0	7.1	0.0	0.0
Hover fly	*Volucella* spp.	0.1	0.0	0.0	0.0	0.0
Hover fly	*Sericomyia* spp.	0.0	0.0	0.2	0.0	0.0
Hover fly	*Syrphus* spp.	0.0	0.0	2.6	0.0	0.0
House fly	*Helina* spp.	0.6	6.8	0.0	3.2	0.0
House fly	*Musca* spp.	6.2	0.0	0.0	0.0	0.0
House fly	*Phaonia* spp.	0.0	0.1	0.0	0.0	0.0
House fly	*Polietes* spp.	0.0	0.0	0.0	0.0	0.2
Fruit fly	*Drosophila suzukii*	0.0	0.0	0.0	0.5	0.0
Noon fly	*Mesembrina meridiana*	0.0	0.1	0.0	0.0	0.0
Dung fly	*Scathophaga* spp.	5.5	0.0	0.0	0.0	16.8
Flesh fly	*Sarcophaga* spp.	1.0	0.0	0.0	0.2	0.1
Fly	*Dryomyza anilis*	0.0	23.2	0.0	0.0	0.0
Soldier fly	*Sargus bipunctatus*	0.0	0.1	0.0	0.0	0.0
Tachinid fly	*Tachinidae* spp.	10.4	0.0	0.6	0.0	0.0
Spider	*Araneus diadematus*	0.0	10.6	0.0	0.5	35.1
Spider	*Metellina segmentata*	0.0	0.6	0.0	0.0	0.0
Spider	*Zygiella* spp.	0.0	0.0	0.1	2.8	0.0
Grasshopper	*Acrididae* spp.	0.0	0.0	0.0	0.0	0.1
Mosquito	*Ochlerotatus detritus*	0.0	0.0	0.0	0.7	0.0
Moth	*Phlogophora meticulosa*	0.0	0.0	7.4	0.0	0.0
Algae	*Monodus* spp.	0.0	0.0	0.0	1.9	0.0
Woodlouse	*Oniscus asellus*	0.0	0.2	0.0	0.0	0.0
Hedgehog	*Erinaceus europaeus*	0.0	0.1	0.0	0.0	0.0

## Data Availability

Sequence data has been uploaded to NCBI, submission number SUB12396684, and can be found here: https://submit.ncbi.nlm.nih.gov/subs/sra/SUB12396684. (accessed on 31 January 2023).

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
