# Peer review of "Molecular Identification of Asian Hornet *Vespa velutina nigrithorax* Prey from Larval Gut Contents: A Promising Method to Study the Diet of an Invasive Pest"

_animals, 2023, doi:10.3390/ani13030511_

Round 1
Reviewer 1 Report
Dear Authors,
You presented an interesting solution to the problem of establishing the taxonomic composition of hornet prey. Article from the first lines seems interesting. But you need to work on the structure of the sections of the article (materials and methods, results and discussions). For example, subtitles 3 and 4 in M&M can be combined under the general name "metabarcoding". You can think about the name of the second subheading. You have "sampling" in the second subheading, which repeats the name of the first subheading.
In the Discussion, some subchapters can be moved to the results section (“Prey result”, “Honey bee”, maybe “Low abundance and vulnerable species”).
And some notes:
- Specify the postal code in the affiliation
- Italicize species names on lines 25, 251 and 254.
- Line 82: LeFort et al. missed the point
- Line 303: Maybe “Information about all taxa found in our samples is presented in the table S2”? Because when the reader sees " raw data on all reads" in a sentence, he expects to see the fasta file of the sequences.
- line 345: honey beets
- Check the bibliography. For example, in lines 524 and 535 the name of the journal is in different cases. Line 474: 2021; See how to properly format the reference for this journal.
- The table S2 does not have a table name. Format the supplementary table headers uniformly. That is, a table S1 and a table S2, or Supplementary table 1 and Supplementary table 2
Author Response
Article from the first lines seems interesting. But you need to work on the structure of the sections of the article (materials and methods, results and discussions). For example, subtitles 3 and 4 in M&M can be combined under the general name "metabarcoding". You can think about the name of the second subheading. You have "sampling" in the second subheading, which repeats the name of the first subheading.
Thanks for taking the time to review our manuscript.
The suggested changes to the subheadings have been made: Subheadings on Metabarcoding have been combined. Subheadings of “Samples” and “Sampling…” have been modified to make it clear one is primary sampling of hornets, the other subsampling for gut contents form collection of hornets.
In the Discussion, some subchapters can be moved to the results section (“Prey result”, “Honey bee”, maybe “Low abundance and vulnerable species”).
Some of the text has been moved to the results section but a discussion of these has been retained in the Discussion section. There is a specific interest in impacts on apiculture an biodiversity from the Asian Hornet.
And some notes:
- Specify the postal code in the affiliation
Done
- Italicize species names on lines 25, 251 and 254.
Done
- Line 82: LeFort et al. missed the point
Done
- Line 303: Maybe “Information about all taxa found in our samples is presented in the table S2”? Because when the reader sees " raw data on all reads" in a sentence, he expects to see the fasta file of the sequences.
Done L323– “Filtered read count data…”
- line 345: honey beets
Oops. Done.
- Check the bibliography. For example, in lines 524 and 535 the name of the journal is in different cases. Line 474: 2021; See how to properly format the reference for this journal.
We have carefully checked through the bibliography.
- The table S2 does not have a table name. Format the supplementary table headers uniformly. That is, a table S1 and a table S2, or Supplementary table 1 and Supplementary table 2
Done – Settled on S1 and S2 format.
Table S2 has the header “Filtered relative abundance read count matrix”. This data is not raw data as we manually correct some assignments based on the methods outlined – the updated header should now reflect this.
Reviewer 2 Report
This manuscript can be accepted after more careful language check and revisions.
Author Response
Thanks for taking the time to review our manuscript. We have further rechecked and proofed of the text.
Reviewer 3 Report
Dear Editor, thank you very much for your invitation to review Manuscript ID: animals-2125298, submitted to “Animals”.
The original paper, “The molecular identification of Asian hornet Vespa velutina nigrithorax prey from larval gut contents: a promising method for insights into the dietary preferences of an insectivorous invasive pest” by Kirsty Stainton et al., provides valuable information about the invasive wasp Vespa velutina nigrithorax (Hymenoptera: Vespidae) focusing in a new molecular approach using dissected larvae from destroyed nests. This matter is exciting and deserves publication after the authors make a few improvements. Please, see all my comments below:
- Check out all the subsections.
Title: It is good. However, it could be shorter.
“The molecular identification of Asian hornet prey from larval gut contents: a promising method for insights into the dietary preferences of an insectivorous invasive pest” or bind the sentences.
Simple Summary: It is well described.
Abstract: It is well described.
- How many samples did you use per nest?
- L25: “V. velutina”
Keywords: “Apis mellifera”, “honey bees”. The authors could change one of these words.
Introduction:
- L59 - 63: “By studying the diet of V. v. nigrithorax, a better understanding of the potential impact it may have on the local fauna and on beekeeping can be ascertained, as well as a better understanding of its natural history to help guide management by: a) determining the optimal location to detect foraging hornets and b) to create management plans to better protect biodiversity and apiculture as necessary.” Can you provide any reference?
- L88: “Vespa velutina nigrithorax” to “V. v. nigrithorax”
- L88-96: Perhaps, you can add this paragraph before your purpose/goals.
2. Materials and Methods:
- Why did you select/choose only five nests? It is a limitation of the current version. Could you explain why you have samples from different years? It needs to be clarified.
- L115: “2.1. Samples”
- L140: “2.2. Sampling, DNA extraction, and PCR”
- L141-142: “Gut contents were carefully dissected from the largest larval stages into Eppendorf tubes for each individual, taking care to avoid any larval gut tissue.”. Did you dissect the larvae with any buffer?
“2.2. Sampling, DNA extraction, and PCR”: The authors could add subsections to explain the procedure for each nest.
- L162: “2.3. Jersey sample metabarcoding”
- L176: “2.4. Tetbury, Woolacombe, Gosport and Alderney metabarcoding”
- L186: “2.5. Bioinformatics Analysis”
- The outcomes are fascinating. However, this section needs to be better supported based on the substantial limitations of methods (mainly the sample number and the problems with some samples).
- The purpose of this manuscript is exciting and relevant, but there are gaps at the present stage that must be solved before publication.
Author Response
Thank you very much for taking the time to review our manuscript and for your suggestions for improvement. We have made our responses in italics, and have pasted your comments and our responses below.
- Check out all the subsections.
We have revised and numbered subsections
Title: It is good. However, it could be shorter.
“The molecular identification of Asian hornet prey from larval gut contents: a promising method for insights into the dietary preferences of an insectivorous invasive pest” or bind the sentences.
It is rather long, we have revised to: “Molecular identification of Asian hornet prey from larval gut contents: a promising method to study the diet of an invasive pest”
Simple Summary: It is well described.
Thanks.
Abstract: It is well described.
- How many samples did you use per nest?
Added to abstract
- L25: “V. velutina”
Done
Keywords: “Apis mellifera”, “honey bees”. The authors could change one of these words.
Have changed “honey bees” to “apiculture”.
Introduction:
- L59 - 63: “By studying the diet of V. v. nigrithorax, a better understanding of the potential impact it may have on the local fauna and on beekeeping can be ascertained, as well as a better understanding of its natural history to help guide management by: a) determining the optimal location to detect foraging hornets and b) to create management plans to better protect biodiversity and apiculture as necessary.” Can you provide any reference?
We have not found sufficiently similar refences to cite here (there are questionably relevant studies on the diets of invasive vertebrates). a) is an outcome we discussed with the end users of the data (Bee Health Inspectors). This is difficult to capture in the text. B) is an outcome we inferred that the data will be used for from knowledge of apiculture.
- L88: “Vespa velutina nigrithorax” to “V. v. nigrithorax”
Done
- L88-96: Perhaps, you can add this paragraph before your purpose/goals.
We have moved this to just above the final paragraph in the introduction.
Materials and Methods:
- Why did you select/choose only five nests? It is a limitation of the current version.
Restrictions on funding and uncertainty whether the method would work restricted the number of nests examined. However, this small number also reflects the low number of nests (with larvae) found and stored in the UK.
Could you explain why you have samples from different years? It needs to be clarified.
Samples are from different years as these were the only nests found in the UK in those years. The Asian hornet has not established yet in the UK, however we had not discussed this in the introduction. We have inserted an appropriate line the introduction, paragraph 1, line 44. The maximum number of colonies with larvae found in the UK in a single year so far has been two. For this reason, it was not possible to only have samples from a single year.
- L115: “2.1. Samples”
This has been modified to “2.1 Collection of hornet samples”
- L140: “2.2. Sampling, DNA extraction, and PCR”
This has been modified to “2.2 Sampling of gut contents, DNA extraction and PCR”
- L141-142: “Gut contents were carefully dissected from the largest larval stages into Eppendorf tubes for each individual, taking care to avoid any larval gut tissue.”. Did you dissect the larvae with any buffer?
No, samples were taken straight through to extraction.
“2.2. Sampling, DNA extraction, and PCR”: The authors could add subsections to explain the procedure for each nest.
- L162: “2.3. Jersey sample metabarcoding”
- L176: “2.4. Tetbury, Woolacombe, Gosport and Alderney metabarcoding”
These two subheadings have been combined to a single numbered subheading “ 2.3 Metabarcoding.”
- L186: “2.5. Bioinformatics Analysis”
Heading now numbered
- The outcomes are fascinating. However, this section needs to be better supported based on the substantial limitations of methods (mainly the sample number and the problems with some samples).
We accept the limitations of the results and feel we have added sufficient caveats to the interpretation of the data already. In the Abstract, we describe our results as a ‘snapshot’, making it clear that the data are small, and that further large scale studies would be necessary “These results […] give a first snapshot of the prey items captured by V. v. nigrithorax in the UK. This method could be used for future large-scale testing from the gut contents of hornet nests, to provide a greater insight into the foraging behaviour of this predator across Europe and elsewhere”.
The final sentence in the Introduction (L. 115) also discusses the limitations: “The results from the study are small scale (a relatively small number of samples, from five nests) and are unlikely to be fully representative of the hornet’s diet should it establish in the UK. With those caveats in mind, we explore the implications of the findings for UK apiculture and biodiversity.” [Slightly modified from original submission]
Within the Discussion we also have a paragraph outlining the limitations of the study and urging caution in the interpretation of the results (fourth paragraph in the 4.1, Line 342 -360), and add caveats in other sections as well (final sentence of section 4.3, L. 411; second paragraph in section 4.4 L. 422 highlighting the limited number of samples, the potential ‘oddness’ of the nests, and restricted temporal range.
- The purpose of this manuscript is exciting and relevant, but there are gaps at the present stage that must be solved before publication.
We have addressed the issues and gaps that have been presented to us, we hope to the reviewer’s satisfaction.
Reviewer 4 Report
The authors explored the species composition of the diet of the asian hornet Vespa velutina nigrithorax from 5 location in UK. To do so, they sampled larvae from nests and they analyzed their gut contents thanks to metabarcoding technique. They detected different taxa in each larval sample with the species composition variation between individual and nest. Wasps, spiders, honey bees and blow flies were the most abundant taxa.
Analysis larvae gut content through metabarcoding seems to be a promising technique to easily investigate species composition diet of invasive insect such as the Asian hornet. Despite the novelty of this approach, my major concern about this experiment is that methodology need to be improved to be really conclusive. Authors pointed out different limitations in their material and methods section and they specified that results need to be considered with cautious. In addition, my feelings is that results are too descriptive (e.g. no statistical analysis).
I also have some minor revisions:
Line 25: “V. velutina” must be in Italic
Lines 116-123: I would appreciate to have GPS coordinates about the different locations. Please also add scale on Figure 1
Lines 122-123: I would delete this sentence
Lines 439-442: Maybe I misunderstood your idea, but you mentioned in your conclusion that your method could help for triangulation and destruction of nests. If your metabarcoding technique relies on larvae gut content, it means that you already know where the nests are.
Author Response
Thank you very much for taking the time to review our manuscript and for your suggestions for improvement. We have made our responses in italics, and have pasted your comments and our responses below.
Analysis larvae gut content through metabarcoding seems to be a promising technique to easily investigate species composition diet of invasive insect such as the Asian hornet. Despite the novelty of this approach, my major concern about this experiment is that methodology need to be improved to be really conclusive. Authors pointed out different limitations in their material and methods section and they specified that results need to be considered with cautious. In addition, my feelings is that results are too descriptive (e.g. no statistical analysis).
We accept the limitations of the results and feel we have added sufficient caveats to the interpretation of the data already. In the Abstract, we describe our results as a ‘snapshot’, making it clear that the data are small, and that further large scale studies would be necessary “These results […] give a first snapshot of the prey items captured by V. v. nigrithorax in the UK. This method could be used for future large-scale testing from the gut contents of hornet nests, to provide a greater insight into the foraging behaviour of this predator across Europe and elsewhere”.
The final sentence in the Introduction (L. 115) also discusses the limitations: “The results from the study are small scale (a relatively small number of samples, from five nests) and are unlikely to be fully representative of the hornet’s diet should it establish in the UK. With those caveats in mind, we explore the implications of the findings for UK apiculture and biodiversity.” [Slightly modified from original submission]
Within the Discussion we also have a paragraph outlining the limitations of the study and urging caution in the interpretation of the results (fourth paragraph in the 4.1, Line 342 -360), and add caveats in other sections as well (final sentence of section 4.3, L. 411; second paragraph in section 4.4 L. 422 highlighting the limited number of samples, the potential ‘oddness’ of the nests, and restricted temporal range.
In light of the limitations of the data, we felt that descriptive summaries were more appropriate than statistical analyses of the data.
I also have some minor revisions:
Line 25: “V. velutina” must be in Italic
Done
Lines 116-123: I would appreciate to have GPS coordinates about the different locations.
We have added approximate Lat / long co-ordinates for the sampling sites, at low resolution to preserve landowner anonymity.
Please also add scale on Figure 1
Done
Lines 122-123: I would delete this sentence
Although it is slightly repetitively worded, we feel it is important to describe the transportation and handling methods to other researchers who may want to do a similar study, and will keep this in.
Lines 439-442: Maybe I misunderstood your idea, but you mentioned in your conclusion that your method could help for triangulation and destruction of nests. If your metabarcoding technique relies on larvae gut content, it means that you already know where the nests are.
The paragraph is perhaps clumsily worded; information about the feeding habits of the hornets can be used to help find foraging hornets, which can then be used to triangulate the nest location. We have reworded the paragraph for clarity (L. 439): “Current management efforts in the UK rely on rapidly locating and removing hornet nests, where the presence of hornets trigger search efforts, and the flight lines of foraging hornets are used to triangulate the nest location. Linking the gut contents data with the location of the nest and the time of year across a large number of nests would allow us to gain a better understanding of hornet foraging habits and better guide efforts to find hornets in the landscape. Therefore, defining their favoured prey choice at different times of year has a practical application for pest management.”
Round 2
Reviewer 3 Report
The authors made excellent improvements to the manuscript, which is suitable for publication.
Reviewer 4 Report
Authors have addressed all my comments and have improved their manuscript. I therefore recommend it for acceptance